# Review on the Battery Model and SOC Estimation Method

**Wenlu Zhou, Yanping Zheng \***, **Zhengjun Pan and Qiang Lu**

College of Automobile and Traffic Engineering, Nanjing Forestry University, Nanjing 210037, China; Wenlulu1997_1@163.com (W.Z.); qq1137656476@163.com (Z.P.); lqared@163.com (Q.L.)

\* Correspondence: zhengyp@njfu.com.cn; Tel.: +86-138-5186-4173

**Abstract:** The accuracy of the power battery model and SOC estimation directly affects the vehicle energy management control strategy and the performance of the electric vehicle, which is of great significance to the efficient management of the battery and the improvement of the reliability of the vehicle. Based on the research of domestic and foreign battery models and the previous results of SOC estimation, this paper classifies power battery models into electrochemical mechanism models, equivalent circuit models and data-driven models. This paper analyzes the advantages and disadvantages of various battery models and current research progress. According to the choice of battery model, the previous research results of the power battery SOC estimation method are divided into three categories: the direct measurement method not based on battery model, the estimation method using black box battery model, and the battery model SOC estimation method based on state space. This paper will summarize and analyze the principles, applicable scenarios and research progress of the three categories of estimation algorithms aiming to provide references for future in-depth research. Finally, in view of the shortcomings of the battery model and estimation algorithm of the existing method, the future improvement direction is proposed.

**Keywords:** power battery; battery model; SOC estimation method; research review

## 1. Introduction

Power battery SOC estimation is one of the key technologies of electric vehicles, and its accuracy directly affects the vehicle energy management control strategy and the performance of the electric vehicle, which in turn affects the reliability and cost of the vehicle. It is also an important parameter in the battery management system. On the one hand, it can provide drivers with important information about the driving range. On the other hand, it also provides an important basis for preventing battery overcharge and overdischarge from reducing battery life and battery pack management and maintenance [1]. However, due to the complex electrochemical characteristics of the battery, it exhibits a high degree of nonlinearity during use. The battery SOC state variable cannot be directly measured. It can only be estimated by externally measurable battery terminal voltage, charge and discharge current, etc. In addition, the estimation process is easily affected by factors such as temperature, cycle times, discharge rate, voltage, noise, etc., which makes it difficult to accurately estimate the battery SOC in real time [2]. Therefore, the SOC estimation of the power battery needs to establish an appropriate battery model for research. An accurate and appropriate power battery model can effectively reflect the correspondence between the external parameters of the battery and the internal state of the battery, and simplify and specify the SOC estimation problem. It is very important for the simulation, design and optimization of electric vehicles. The complexity of the model and the computational cost of the processor also affects the decision making and control of the BMS system [3]. It can be seen that the establishment of an accurate and simple battery model and accurate battery SOC estimation directly affects the vehicle energy management control strategy and the performance of electric vehicles.

At present, domestic and foreign researchers have achieved some important results in the preliminary research on battery models and battery SOC estimation. There is a lot of related literature on power battery SOC estimation. These review documents have a certain reference value for the research progress of SOC estimation, but there are many defects with incomplete summary and the lack of process expression, as shown in Table 1. Compared with other review literature, this article summarizes the latest research results, and comprehensively and thoroughly analyzes each battery model and SOC estimation method. The advantages and disadvantages and research progress of each model and estimation method are described in detail. According to the different modeling methods, the battery model is divided into three categories: the electrochemical mechanism model, the equivalent circuit model and the data-driven model. The article specifically analyzes the main characteristics and development trends of each type of model, and comprehensively analyzes and compares the advantages and disadvantages of various models. According to the choice of battery model, the preliminary research results of power battery SOC estimation are divided into three categories: the direct measurement method not based on battery model, the estimation method based on black box battery model, and the estimation method based on state space battery model. The article systematically sorts out various algorithms and compares their advantages and disadvantages. Finally, the future research trends of power battery models and SOC estimation methods are discussed and prospected.

**Table 1.** Comparison chart of published reviews.

| Review Article | Merits | Demerits |
|:---:|:---:|:---:|
| [4] | In-depth overview of battery SOC estimation methods, focusing on estimation errors and their advantages and disadvantages | Mathematical expressions, flowcharts and structural diagrams of related algorithms are not provided |
| [5] | Focus on summarizing commonly used lithium-ion battery SOC estimation methods, and analyzing the advantages and disadvantages of various methods | The analysis of SOC estimation algorithm and research progress is not comprehensive |
| [6] | The SOC estimation methods of batteries are reviewed, and three battery models and model-based estimation methods are mainly introduced | The data-driven SOC estimation method was not specifically introduced |
| [7] | The SOC estimation method based on the equivalent circuit model is systematically sorted out and compared with advantages and disadvantages. It also introduces in detail the factors affecting the estimation error and its countermeasures | Only the model-based SOC estimation methods are reviewed |
| [8] | It focuses on analyzing the main characteristics of five types of estimation algorithms and comprehensively comparing and discussing the advantages and disadvantages of models and algorithms | The introduction to the battery model is relatively brief |
| [9] | Analyze the improvement of the battery model and the refinement of the algorithm while considering the temperature | The analysis of the research status is not comprehensive enough |

## 2. Battery Model

In terms of battery model research, the battery model required to be established has a good consistency with the external characteristics of the battery. The internal chemical reaction of the battery is a complex non-linear process. The battery is polarized at the moment when the charging and discharging current changes, that is, the battery terminal voltage does not show pure resistance characteristics, but continuously changes in a non-linear manner. The polarization of the battery will cause the resistance of the charging and discharging current to flow through the battery to increase [10]. After long-term use, the battery will still have aging problems, such as battery capacity degradation and internal resistance increase, which will cause the power battery's state of charge to seriously deviate from the true situation [11]. There are individual differences between the batteries of different monomers, from the battery monomer to the battery module to the battery pack, the power performance is significantly attenuated. These factors make it difficult for people to build an accurate battery model to accurately describe all battery performance. People can only use many methods to simulate the characteristics of the battery from different angles. Currently, the commonly used battery models include electrochemical mechanism models, equivalent circuit models, and data-driven models.

### 2.1. Electrochemical Mechanism Model

The electrochemical mechanism model is to establish electrochemical power and transmission equations according to the internal mechanism of the battery, consider the physical and chemical properties of the positive and negative materials, the internal diffusion process of the battery, the electrochemical reaction process, etc., and fully and accurately describe the internal physical and chemical processes and external characteristics of the battery. Electrochemical models mainly include pseudo-two-dimensional models (P2D) [12], single-particle models (SP) [13] and other simplified pseudo-two-dimensional models (SP2D). The P2D model established by M. Doyle, T. F. Fuller and J. Newman based on porous electrode theory, concentrated solution theory, and dynamic equations laid the foundation for the development of electrochemical mechanism models. The P2D model is rigorous and accurate, but its partial differential equations have no analytical solutions. Generally, the finite difference method is used to solve them, which takes a long time, and the coupling between the control equations is high, and the amount of calculation is large, which cannot be applied to real-time SOC estimation. Reference [14] proposes a P2D model parameter identification method based on a heuristic algorithm based on the P2D model of lithium-ion batteries, which effectively reduces the model parameters to be identified and reduces the calculation time. Most electrochemical models are derived and developed on the basis of the P2D model. The SP model is the most mature simplified model based on the P2D model to study the main performance of the electrode and the influence of solid phase diffusion. This model replaces an electrode with a single particle, ignoring the influence of liquid phase concentration and liquid phase potential on the terminal voltage, and consists of only two control equations, reducing the parameters to be identified and improving the calculation and simulation speed. However, due to ignoring too many factors in the simplification process, the accuracy of the model is low, and it is only suitable for small magnification and constant current conditions. Once the current magnification increases, the electrolyte concentration changes significantly, and the model error increases [15]. In order to adapt the SP model to large-rate constant current and dynamic conditions, the solid-phase and liquid-phase lithium-ion diffusion equations were simplified by the three-parameter parabolic method and Padé approximation method, and a simplified reduced-order extended single-particle model (ESP) was established based on the SP model [16–18]. The ESP model can not only ensure the accuracy of the SP model, but also improve the model's adaptability to high current conditions. In order to meet the requirements of different discharge rates and real-time estimation of SOC, the P2D model is reasonably simplified to obtain other SP2D models [19–21]. Reference [20] proposes a simplified multi-particle model using predictor-correction strategy and alignment in order

to maintain the accuracy of the SP model while reducing the complexity of the model. The predictor-corrector strategy is used to solve the approximate value of the electrolyte concentration to reduce the complexity of calculation, and the alignment is used to predict the uneven effect of the electrochemical reaction, and improve the calculation efficiency and estimation accuracy. However, there are still many SP2D model parameters obtained after the simplification, and the sensitivity of each parameter to the output voltage is different, and it is impossible to accurately identify all the parameters. Reference [21] uses the non-linear least squares method combined with the Fisher information matrix to analyze the identifiability of SP2D model parameters on the basis of SP2D model, thereby establishing the SP2D-Iden model and improving the accuracy of SOC estimation.

### 2.2. Equivalent Circuit Model

The equivalent circuit model uses circuit components such as resistors, capacitors, and constant voltage sources to form a circuit network to simulate the dynamic characteristics of the battery. In order to accurately estimate the SOC value of the battery, the model is required to better reflect the static and dynamic characteristics of the battery. However, the order of the model should not be too high, which reduces the amount of processor calculations and is easy to implement in engineering [22]. In addition, from an electrochemical point of view, the selected equivalent circuit model should reflect the relationship between the battery's electrochemical reaction process, electrode solids concentration, electrolyte concentration, and open circuit voltage. Equivalent circuit models are divided into integer-order models and fractional-order models. Common integer-order equivalent circuit models include the Rint model, Thevenin model, PNGV model and multi-order model, as shown in Figure 1.

The Rint model [23] uses an ideal voltage source $U_{oc}$ and the battery DC internal resistance $R_0$ in the series to describe the dynamic characteristics of the power battery. $R_0$ and $U_{oc}$ are functions of SOC and temperature. This model has a simple structure and is easy to implement, but the model has low accuracy and fails to describe the polarization phenomenon inside the power battery. It is an ideal situation and is not suitable for applications in electric vehicles. The Thevenin model [24] is based on the Rint model, adding a parallel RC network to simulate the polarization effect of the battery. This model has a relatively simple structure and high simulation accuracy. It can also describe the polarization effect inside the power battery. When the battery is charging or discharging, the change of the voltage at both ends shows both abrupt and gradual change. In the Thevenin model, $R_0$ is used to simulate the abrupt resistance characteristics, and $R_P$ and $C_P$ are used to simulate the capacitance characteristics of voltage gradual changes. Compared with the Rint model, the Thevenin model increases the research on the polarization characteristics of the power battery, which can better simulate the dynamic and static characteristics of the battery. In addition, the model parameters are relatively small, and the curve fitting is mostly a single exponential model. The subsequent estimation process requires less calculation, which is suitable for the SOC estimation of embedded systems and meets the application requirements of electric vehicles [25]. On the basis of the Thevenin model, a capacitor $C_b$ can be connected in the series to form a PNGV model [26]. This capacitor is used to describe the change in battery open circuit voltage caused by current integration during the battery's long-term charging and discharging process. The PNGV model is a typical nonlinear equivalent circuit model, which has high accuracy in simulating the transient response process, and is suitable for large current, step-type, and more complex charging and discharging conditions. In theory, the model should be more in line with the behavior of the battery in actual work, but the existing equipment cannot detect the polarization process of the battery in detail, so it is impossible to obtain a more accurate capacitance $C_b$ value. In addition, the model has a relatively high complexity, a large amount of calculation, and low real-time performance. In order to better reflect the dynamic characteristics of the ternary lithium battery in the step-type charging and discharging conditions, reference [27] extends the polarization circuit of the

PNGV model and uses a dual RC circuit to replace the original single RC circuit. It more closely characterizes the polarization characteristics of the battery and better simulates the static circuit. Reference [28] proposed a PNGV battery model based on variable parameters, and set $R_0$, $R_p$, and $C_p$ as variable parameters that vary with the battery SOC to reduce the complexity of the model and reduce the amount of calculation. The model can simulate the dynamic working characteristics of the battery in real time, which improves the accuracy of the model. At present, most of the integer-order equivalent circuit models are based on the Thevenin model by adding different circuit elements to obtain better performance. Considering the electrochemical polarization reaction and concentration polarization reaction inside the battery, adding an RC network to become a second-order RC equivalent circuit can further improve the ability of the equivalent circuit to simulate the dynamic characteristics of the battery, thereby improving the estimation accuracy of the battery SOC [29–31]. The multi-level RC equivalent circuit model usually contains more than two sets of RC polarization parameters, which are used to describe the dynamic and static characteristics of the battery. The more RC components, the higher the estimation accuracy of the power battery SOC, but the parameter identification will be more difficult. With the increase in state dimensions and over-fitting problems, calculations will become more complicated. The battery model must simultaneously meet the requirements for accurately capturing the dynamic characteristics of the battery in terms of accuracy and adapting to the real-time performance of the system in terms of complexity. Many researchers put forward a third-order equivalent circuit model on the basis of taking into account accuracy, complexity and practical value. The structure of the model is moderately complex and has high accuracy, which can well reflect the dynamic polarization impedance of the battery and simulate the real-time operating characteristics of the battery [32,33]. Reference [34] combines the Thevenin model, second-order RC model and third-order RC model as a hybrid model describing battery characteristics. The weights of the three models in the hybrid model are calculated using the Bayesian method. However, this method requires a large amount of calculation and is more difficult to use in engineering. If the requirements of accuracy and system reliability are considered at the same time, it is more appropriate to use the Thevenin model and the second-order RC circuit model as the equivalent circuit model of the battery [35]. At present, a large number of battery modeling studies have shown that in practical applications, generally only models below the third order are required to meet the accuracy requirements, and current studies mostly use second-order RC models to build SOC estimators [36–38].

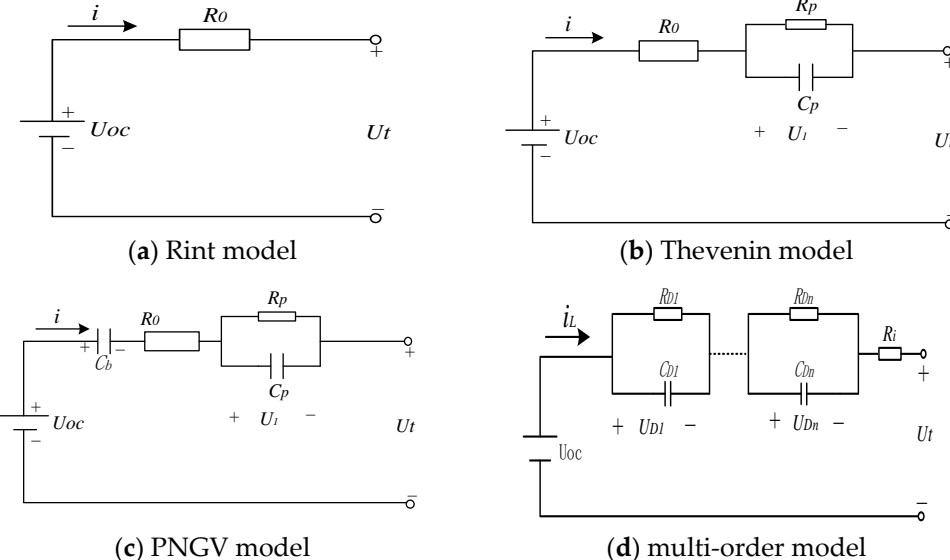

**Figure 1.** Common integer-order equivalent circuit models.

Although the research of the equivalent circuit model based on the fractional order theory started late, it has developed rapidly and has obtained more research results. Figure 2 is a commonly used fractional-order model, replacing the pure capacitive element in the Thevenin model with a constant phase EI-ement (CPE). In impedance spectrum fitting, the CPE is often used in parallel with a pure resistance, and the CPE characteristics are difficult to process in the time domain, which usually need to be processed by the theory of fractional calculus. Commonly used fractional calculus theories have the Grünwald-Letnikov (G-L) definition, Riemann-Liouville (R-L) definition and Caputo definition. For most functions, the definitions of G-L and R-L are equivalent, and G-L provides the most direct form and method for discretization approximation. The definition of Caputo is derived from the definition of R-L. The difference between the two is that Caputo's derivation of constants is bounded, and R-L's derivation of constants is unbounded. Caputo is mainly suitable for the description and discussion of the initial value problems of fractional differential equations, making it more accurate to describe the dynamic characteristics of the power battery terminal voltage than the integer-order equivalent circuit model under the same order. Reference [39] is based on G-L fractional calculus theory, mathematically derives the discrete space state expression of equivalent circuit and establishes a first-order fractional equivalent circuit model, which provides a model basis for power battery SOC estimation. The impedance element in the fractional-order model can more accurately describe the electrochemical process of lithium-ion batteries, such as charge transfer, electric double layer effect, material transfer and diffusion, etc. It not only improves the accuracy, but also effectively solves the computational complexity caused by too many modules [40]. Reference [41] uses a fractional-order model containing a CPE to simulate the voltage curve under different conditions. The results show that its accuracy is higher than that of the integer-order model of different orders. Compared with the traditional equivalent circuit model, the fractional equivalent circuit model has the advantages of high accuracy and flexible calculation. For example, the fractional-order model and fractional-order PNGV model established based on the second-order equivalent circuit model in references [42,43] can more realistically simulate the polarization effect and charge-discharge characteristics of the battery. The selection of a general power battery model must not only meet certain accuracy requirements, but also avoid being too complicated. Reference [44] established three fractional high frequency equivalent circuit models based on electrochemical impedance spectroscopy, which ensured the accuracy of the model and reduced the complexity of the model, providing a reference for the choice of battery model.

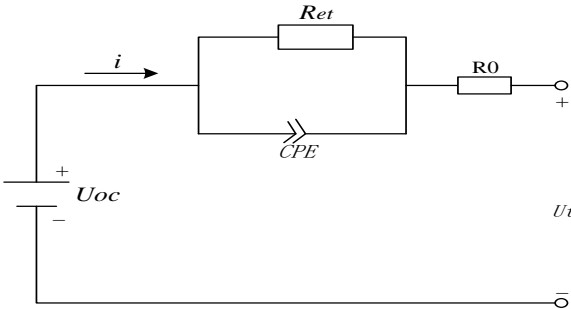

**Figure 2.** A commonly used fractional model.

### 2.3. Data-Driven Models

Data-driven models have received widespread attention due to their flexibility and model-free advantages. It avoids the modeling and parameter identification problems based on model estimation methods, and can directly analyze the hidden information and evolution rules from the external characteristic parameters of the battery. The data-driven method is widely used in battery modeling, has a high degree of non-linearity and self-learning characteristics, and has a good generalization ability for estimating battery SOC in a non-linear system. Data-driven models mainly include neural network

models, autoregressive models, and support vector machine models. In the modeling process of the data-driven model, since there is no clear model structure to simulate the internal reaction of the battery, only sufficient test data can be used to train the data model. It can be applied to various types of batteries regardless of the type of battery [45]. Reference [46] constructed a random forest regression model for SOC estimation, which effectively avoided the problem of over-fitting, improved the estimation accuracy, and provided a reference for future estimation model research. Due to the frequent changes in operating conditions and large differences in energy consumption under different operating conditions, it is difficult to measure electrochemical parameters in the actual driving process of the vehicle. Reference [47] analyzes vehicle energy consumption and extracts energy consumption factors. Based on the collected vehicle operation data, machine learning algorithms such as Lasso, Ridge, LGBoost, XGBoost are used to train the data, and the energy consumption of temperature stratification is proposed. The model has high accuracy and good prediction effect. Reference [48] proposed a radial basis function neural network model to eliminate the impact of battery degradation on the accuracy of the original training model. Although the data-driven model has many advantages, it requires a large amount of battery experimental data as a drive. In the case of a small number of data samples, the estimation accuracy is relatively general and the versatility is poor. Moreover, the implementation of the algorithm takes a long time, and the real-time performance of the application is difficult to guarantee. For the real-time performance of electric vehicles, it is a greater challenge.

## 3. Research on SOC Estimation Algorithm

At present, there is much domestic and foreign research on battery SOC estimation. According to the selection of models in battery SOC estimation, battery SOC estimation algorithms are roughly divided into three categories: direct measurement method not based on battery model, the SOC estimation method using the black box battery model, and the SOC estimation method based on the state space battery model.

### 3.1. Direct Measurement Method Not Based on Battery Model

The direct measurement method that is not based on the battery model is to estimate the battery SOC based on the battery's voltage, current, internal resistance, impedance and other reproducible battery parameter variables that have a significant correlation with the battery. These battery parameter variables should be relatively easy to measure in actual use. Direct measurement methods that are not based on battery models mainly include the ampere-hour integration method, open circuit voltage method, internal resistance method, impedance spectroscopy method, load voltage method, and special methods suitable for specific objects. The ampere-hour integration method is also known as the coulomb measurement method or the ampere-hour measurement method. Its essence is to estimate the battery SOC by accumulating the amount of electricity charged or discharged when the battery is charged and discharged. This method is simple and straightforward, and has low requirements for controller hardware and storage. It is the basis of many estimation algorithms and is currently the most used method. However, the ampere-hour integral method also has some defects in estimating battery SOC [49]. This method requires high accuracy of current sensors, and the accuracy of current sensors in practical applications will be affected by noise, temperature drift and other random disturbances. Furthermore, with the increase of time, the cumulative error of the battery SOC becomes larger and larger. This method does not have the initial convergence and has a strong dependence on the initial state value, so the accuracy of the initial battery SOC has a greater impact on the estimation accuracy. This method also considers the battery charging and discharging efficiency. In the case of high temperature and severe current fluctuations, the estimation error is relatively large. At the same time, the battery static capacity decline caused by the decline of the power battery performance will also affect the accuracy of the SOC estimation. In order to avoid the constraints of the above factors and improve

the calculation accuracy, the reference [50] proposed ampere-hour integration method with capacity correction. The initial SOC value of the battery pack is obtained through the open circuit voltage method. On the basis of the traditional ampere-hour integration method, the correction factors of the charge and discharge rate, temperature and charge-discharge coulomb efficiency are obtained through experiments to modify the ampere-hour integration method capacity. This method effectively eliminates the capacity error of the traditional ampere-hour integration method, but it does not give a specific model, and it is difficult to realize engineering applications. In order to facilitate the engineering application, the reference [51] determines the model parameters according to the battery discharge data of different ambient temperature and current changes, and thus proposes a new capacity correction model. The ampere-hour integration method using the new model for capacity correction can effectively eliminate the cumulative error in the ampere-hour integration method. At present, the ampere-hour integration method is often used in combination with other algorithms in practical engineering applications, such as the open circuit voltage method [52–54].

The open circuit voltage (OCV) of a battery is the voltage when the battery is in a steady state in an open circuit condition, which is close to but smaller than the electromotive force of the battery in value. The relatively fixed functional relationship between the battery's OCV and the SOC is used to estimate the battery SOC value. The corresponding relationship can be obtained by looking up a table or curve fitting. Generally, the SOC-OCV curves measured at different standing times are slightly different. The longer the standing time, the more accurate the measurement of the OCV. Considering the test efficiency, the OCV can be measured by standing for 1 h in accordance with the requirements of the national standard, which can meet the requirements of the project. The current battery SOC-OCV relationship curve is usually obtained based on experiments using polynomial fitting methods [55,56]. The fitting accuracy becomes higher as the order of the polynomial increases, but the increase of the order will also increase the degree of non-linearity and increase the amount of calculation. Therefore, some scholars use logarithm to fit the SOC-OCV curve. The logarithmic fitting method is applied to the ternary lithium battery with higher fitting accuracy, but the fitting accuracy is lower when applied to the open circuit voltage of the lithium-iron phosphate battery when the SOC is higher or lower [57]. Reference [58] uses a double exponential fitting method in SOC-OCV curve fitting, and adds a square term on the basis of the double exponential function. This fitting method has fewer equation coefficients and higher fitting accuracy. However, there are problems with highly non-linear characteristics and difficult to use in real vehicles. Reference [59] proposes a linear SOC-OCV curve fitting method that can be used in real vehicles. According to the characteristics of the SOC-OCV curve, a piecewise straight-line fitting is performed to reduce the amount of calculation, but the fitting accuracy is reduced. The open circuit voltage method is relatively simple in structure, and has a good SOC estimation effect in the initial and final stages of charging, but it requires a long time for the battery to stand still to achieve voltage stability. Therefore, it is only suitable for electric vehicles in the parking state when used alone. In engineering, the open circuit voltage is often combined with the ampere-hour integration method to correct the cumulative error of the ampere-hour integration method. Its fitting method is used in the controllable voltage source of the equivalent circuit.

The internal resistance method uses the monotonic relationship between the internal resistance of the battery and the SOC to estimate the SOC value of the battery under the condition of knowing the internal resistance of the battery. The internal resistance of the battery is divided into AC internal resistance and DC internal resistance, respectively expressed as the resistance of the current to AC and DC. The AC internal resistance is greatly affected by temperature, so it must be measured with an AC impedance meter. The principle and application of the AC impedance method are detailed in reference [60]. The DC internal resistance is the ratio of the battery voltage change to the current change in the same short period of time. Normally, the battery is charged or discharged with

constant current from the open circuit state, and the difference between the load voltage and the open circuit voltage in the same time divided by the current value is the DC internal resistance. It should be noted that if the time period is shorter than 10 ms, only the ohmic internal resistance can be detected; if the time period is longer, the battery internal resistance will become complicated. So, it will be difficult to accurately measure the battery internal resistance. The battery internal resistance measuring device is expensive and large in size, and the internal resistance of the battery is generally on the order of milliohms, which is easily affected by factors such as temperature and cycle times. Especially in the driving of the car, there is a large electromagnetic interference, which makes it difficult to accurately measure the internal resistance of the battery in the conventional circuit. Therefore, the internal resistance method is not suitable for online estimation of power battery SOC [61]. In engineering applications, the internal resistance method is often used in combination with other algorithms [61–63].

The discharge test method is the most accurate and reliable method for determining the SOC of a power battery. It is suitable for any battery. The current remaining capacity can be obtained by the discharge test method. The discharge test method is an experimental method in which the power battery is continuously discharged at a certain discharge rate (usually 0.3C or 1C) at a constant current until the battery terminal voltage reaches the discharge cut-off voltage. The integral of the discharged current value over time is used as the SOC value of the battery [64]. The discharge test method is often used in the laboratory as a reference standard for battery capacity testing. It is the most reliable SOC estimation method and is applicable to all types of batteries. However, it also has shortcomings: the test takes a long time, only after the entire discharge test is over, the SOC value at each time can be calculated, and real-time estimation of SOC cannot be achieved; strict test conditions are required, constant current and accurate measurement are required; during the test, the battery in operation must be terminated and switched to a constant current discharge state. Therefore, the discharge test method is not suitable for driving electric vehicles, and can be used for the maintenance of power batteries and the identification of battery model parameters.

Electrochemical impedance spectroscopy (EIS) is an important method for studying the interface reaction mechanism and electrode process of the electrode and the electrolyte. It plays an important role in establishing the electrochemical mechanism model of lithium-ion batteries for the study of the electrode process. By testing the EIS when the battery is discharged to different SOC values, parameters such as the charge transfer internal resistance, total ohmic internal resistance and Warburg impedance of the battery are obtained. According to the relationship between the obtained parameter and the SOC value, find the parameter that has a monotonous relationship with the SOC, and the SOC value of the battery can be estimated according to the parameter during use [65]. The accuracy of estimating battery SOC based on EIS is high, and it can quickly and non-destructively directly reflect the dynamic characteristics of batteries. However, the battery impedance is costly, greatly affected by battery life, and is sensitive to temperature. When the battery temperature changes greatly, it is difficult to accurately estimate the SOC value. In order to explore the influence of temperature on battery impedance, reference [66] measured the EIS of lithium-iron phosphate batteries at different SOCs and different temperatures. Using the information of the whole frequency band, the change of battery EIS was explored from both the amplitude and phase. It was found that the battery impedance phase at a certain frequency has a strong linear relationship with the SOC at a certain temperature, which can be used as a parameter for estimating SOC. Reference [67] research on lithium-iron phosphate battery EIS found that when the temperature of the environment where the battery is fixed, the current SOC value of the battery can be obtained by only measuring the impedance of the battery at a single frequency.

At the moment when the battery starts to discharge, the battery voltage quickly changes from the open circuit voltage state to the load voltage state. If the current is constant, the change rule of the load voltage is similar to the open circuit voltage [68].

Compared with the open circuit voltage method, the load voltage method has a better estimation effect in constant current discharge. However, in practical applications, due to the large changes in the demand current, the load voltage method cannot be used to estimate the battery SOC during driving. It is usually only used to determine whether the discharge is cut off.

A special method is proposed for a certain type of battery. For example, the stable internal pressure method is used to measure the SOC of a nickel-hydrogen battery in reference [69]. After the nickel-hydrogen battery is left standing, its internal stable pressure has a corresponding relationship with the battery SOC. However, this method requires an internal pressure sensor, and whether it can be applied to other types of batteries remains to be studied.

### 3.2. SOC Estimation Method Based on the Black Box Battery Model

The black box battery model regards the battery as an unknown system, takes the online measurable battery current, voltage, temperature, etc., as the input of the model, and the battery SOC as the output of the model. It trains input and output data through some intelligent algorithms, and establishes the relationship between input and output, as shown in Figure 3. The black box battery model usually uses neural networks, support vector machines, fuzzy algorithms, deep learning and other methods to obtain the estimation method of the battery SOC value according to the input battery state parameters.

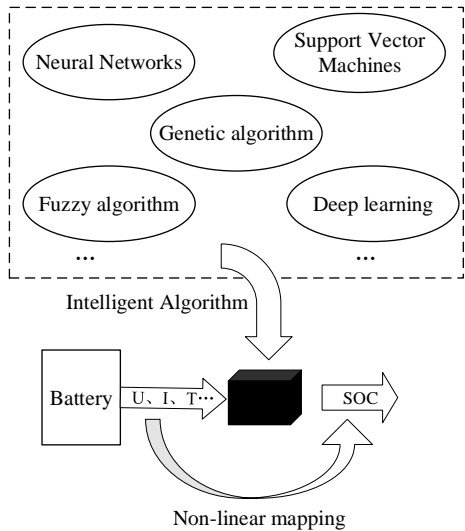

**Figure 3.** SOC estimation method based on black box battery model.

The neural network model is obtained by simulating the network structure of the animal nervous system, and has good adaptability to nonlinear systems. A typical neural network model consists of an input layer, a hidden layer and an output layer, as shown in Figure 4. According to the number of inputs and outputs, the number of nodes in different layers can be defined, and the information can be processed in parallel. It has the characteristics of multiple input and multiple output, fault tolerance, self-learning and wide range of use. It is suitable for various batteries. However, the use of this model requires a lot of reference data for training, and only has good results when processing data within the range of training samples. The battery SOC estimation error is affected by the training data and training methods, which limits its application. Usually, this algorithm does not perform SOC estimation alone. It is often combined with some data clustering algorithms. In the previous research results, research scholars combined neural networks and fuzzy logic, so that they have the ability to imitate the fuzzy reasoning of human thinking. The fuzzy c-means clustering (FCM) algorithm is usually used to divide the input nonlinearly to reduce the number of fuzzy rules and the complexity of the system [70,71],

or an improved method based on this [72]. Some researchers also use neural network algorithms in series with various improved Kalman filters, and use BP neural networks to establish an error compensation model for the extended Kalman filter estimation process to solve the insufficient accuracy of the extended Kalman filter algorithm alone [73–75]. The combination of the neural network and Kalman filter can quickly and accurately estimate the battery SOC value while improving the robustness of random noise and error peaks [76]. Aiming at the difficulty of neural network modeling and optimization, the particle swarm optimization (PSO) algorithm is used to optimize the number of nodes in the hidden layer of the neural network. It can not only avoid local solving problems of the algorithm and reduce the prediction error of the algorithm, but can also improve the generalization ability and practical application ability [77]. Finally, the optimized neural network is used to estimate the battery SOC to solve the difficult problem of neural network modeling. In order to solve the problem of inconsistent estimated values caused by the instability of the initial value of the neural network and network parameter settings, the use of dual neural networks for real-time battery parameter identification and battery SOC estimation can improve the estimation accuracy while reducing the calculation pressure [78]. The neural network algorithm is suitable for various power batteries, but requires a lot of data for training. The result is greatly affected by the size of the sample and the training method, which reduces the practical application ability of the model.

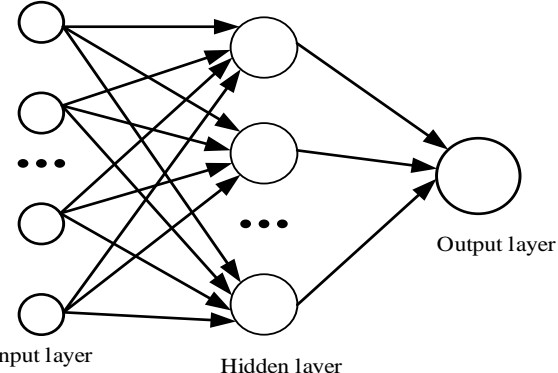

**Figure 4.** Typical neural network model structure.

Support vector machine (SVM) is a more commonly used and mature machine learning algorithm. It seeks to minimize the structured risk to improve the generalization ability of the learning machine, so as to minimize the experience risk and confidence range. Thus, a good statistical law can be obtained even when the number of statistical samples is small. SVM is divided into support vector classification (SVC) for classification problems and support vector regression (SVR) for fitting regression. SVM has good effects in nonlinear and high-dimensional battery modeling. It can accurately estimate the SOC of the battery, but increases the computational complexity. In order to adapt to the non-linear characteristics of lithium-ion batteries, it is necessary to use the kernel function to map the input into high dimensions, and use the quadratic programming method to find the optimal support vector. Aiming at the problem of battery SOC estimation, both training samples and verification samples are a collection of sample points composed of battery state parameters such as voltage, current, temperature, and SOC at a certain moment. By selecting the appropriate kernel function, training the support vector machine model, and obtaining the optimal hyperplane, and using the new sample set to verify the model, it can be judged whether the support vector machine meets the requirements of accuracy and real-time performance. Reference [79] combines the extended Kalman filter with the support vector machine and obtains the SOC estimation value initially from the EKF algorithm, and trains the filtered output data of the EKF algorithm to obtain the SVM model. The regression prediction ability of the obtained SVM model is used to compensate the error of the preliminary SOC estimation value, thereby improving the accuracy of the SOC estimation.

Fuzzy algorithms use computers to imitate human reasoning and decision-making processes, and use fuzzy sentences to operate during the reasoning process. When using the fuzzy algorithm to estimate the battery SOC, the voltage, current, and temperature of the battery need to be fuzzy. The precise value is converted into fuzzy variables that can be identified by the fuzzy control system. Then, based on experience, fuzzy language is used to establish fuzzy rules for fuzzy reasoning. Finally, decision and de-fuzzy processing on the inference result to get the battery SOC value and output it [80]. In the application process, the selection of different variables, membership functions, fuzzy rules, inference algorithms, etc., will have a certain impact on the output results. At present, fuzzy algorithms are often combined with other intelligent algorithms to obtain higher performance. For example, reference [81] proposed a nonlinear correlation fuzzy support vector machine algorithm, and carried out real-vehicle SOC estimation on pure electric vehicles. The experimental results show that the proposed algorithm can enhance the anti-noise ability of the system and improve the measurement accuracy. Another example is the combination of the neural network method mentioned above, both of which adopt a parallel processing structure to obtain the input and output relationship from the input and output samples of the system. Therefore, for the highly nonlinear power battery, the parallel structure and learning ability of the two can be used to estimate the power battery SOC [82]. In order to reduce the impact of current and SOC value on the accuracy of the first-order RC model, reference [83] proposed a fuzzy dual Kalman filter (FDKF) algorithm that dynamically corrects the covariance of the observed noise. The Kalman filter algorithm is used to update the transformed model parameters, the fuzzy control system is established to adjust the covariance value of the observed noise to offset the model error, and finally, the battery SOC value is estimated by the extended Kalman filter.

Deep learning is essentially a neural network with more layers, which can automatically extract more abstract and expressive features from samples, thereby realizing complex nonlinear mapping between input and output data. Deep learning organizes multiple neurons with simple processing capabilities, so that complex nonlinear networks have strong generalization capabilities and parallel processing capabilities. Then the battery voltage, current, temperature and other information are input into the deep learning network input layer, and through the calculation of hidden layer nodes, the output result of the battery SOC is finally obtained. Its training model is more complex, can achieve higher estimation accuracy, and has higher requirements on computing resources and computing time. The algorithms that implement the deep learning theory include the deep belief network (DBN), convolutional neural network (CNN), and recurrent neural network (RNN). DBN consists of n-layer restricted Boltzmann machine network (RBM) and a BP network, as shown in Figure 5a. It can realize the organic combination of unsupervised learning and supervised learning, effectively reduce the training error of the prediction model, and improve the prediction accuracy [84]. A typical CNN structure consists of an input layer, a convolutional layer, a pooling layer, a fully connected layer, and an output layer, as shown in Figure 5b. After multiple filter operations, CNN can extract data features through layer-by-layer convolution and pooling operations. However, there is no interconnection between neurons in each layer of CNN and DBN networks, and the structure of one input corresponding to one output cannot solve the time series problem. RNN is composed of input layer X, hidden layer Y and output layer H. Different from CNN and RNN networks is the delayer that RNN retains historical information [85], as shown in Figure 5c. The RNN is widely used to solve time series data problems. However, RNN has the problem of gradient explosion and gradient disappearance, so it can only deal with shorter timing problems. It is greatly restricted in practical applications. The research of gated recurrent unit (GRU) and long-short term memory (LSTM) networks can effectively improve the hidden nodes of RNN and provide a new direction for solving the problem of time series prediction. As a variant of the RNN network model, the LSTM network model can well solve the defects of the original RNN by introducing the unit state. It is more suitable for dealing with and predicting relatively long intervals and

delays in time series. Combining LSTM and CNN can make full use of the input data features while saving and saving historical input information, which has a more accurate and stable prediction effect establishment [86]. GRU is a variant of LSTM, which is used to overcome the short-term dependency problem of simple RNN. It has strong robustness when the initial SoC value is uncertain, and can adapt to changes in ambient temperature well [87]. The GRU-RNN can self-learn network parameters by adaptive gradient descent algorithms. Compared with electrochemical models and equivalent circuit models that contain differential equations, the GRU-RNN is free from requiring a large amount of work to hand-engineer and parameterize [88]. Compared with LSTM-RNN, GRU-RNN uses a simpler structure and fewer parameters, and is better than LSTM-RNN on a smaller data set.

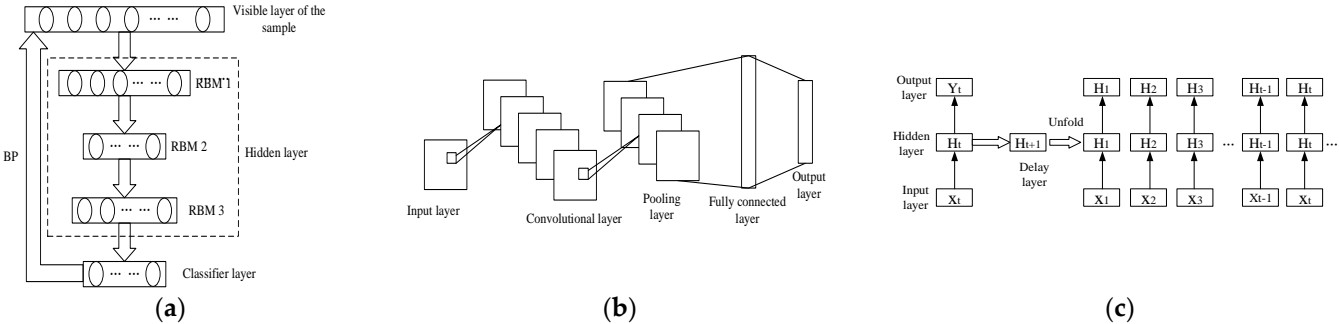

**Figure 5. Deep learning structure**: (**a**) structure of DBN; (**b**) structure of typical CNN; (**c**) structure of RNN.

Genetic algorithm (GA) is an intelligent optimization method for solving constrained and unconstrained, stochastic and nonlinear problems with the continuous development of optimization theory. In terms of computational operations, the GA has a high degree of parallelism, which can be used in parallel in generating offspring and calculating individual fitness values. It has the ability to self-organize, self-adapt, self-learn, and group evolution. Additionally, it has the characteristics of implicit parallelism and searchability of the global solution space [89]. The most important part of the GA is the selection and establishment of fitness function. The fitness function can be any function, and there is no special standard restriction. Because the evaluation of the individual's pros and cons in the algorithm depends only on the fitness function. Therefore, the choice of the fitness function has a profound impact on the process and results of the genetic algorithm. Compared with the traditional identification method, the GA is more robust. It only needs the value of the objective function to randomly select the optimal parameters that meet the conditions. It has certain feasibility and efficiency for finding the optimal EKF noise matrix. Reference [90] introduces GA to online optimization of the covariance of the system noise matrix and measurement matrix in EKF, so as to realize the online estimation of battery SOC when the model error is the smallest. The GA has high complexity and slow global search speed, but when encountering multiple extreme values, it is easy to fall into the local optimum. It can better jump out of the local optimum through selection, crossover, mutation, etc., so as to perform a global search to find the global optimum. The multi-algorithm collaborative optimization intelligent identification method that combines the PSO algorithm that is easy to fall into the local optimum and the GA can quickly capture the search range of the feasible solution space, realize the fast search for the optimal solution of the battery model parameter identification problem, and the identification accuracy is high [91]. The optimization method of the lithium-ion equivalent circuit model based on the GA algorithm can accurately characterize the high dynamics of the lithium-ion battery [92]. The GA is also often combined with the neural network algorithm to estimate the power battery SOC. Reference [93] proposed a novel immune genetic algorithm (IGA) and BP neural network combined power battery SOC estimation method. The IGA is used to optimize the

parameters of the BP neural network, and the feasibility and effectiveness of this algorithm are verified through simulation and experiments under battery conditions.

At present, domestic and foreign research on neural networks, support vector machines, deep learning, genetic algorithms, etc., were conducted, and the accuracy of SOC estimation methods was also improved. However, the SOC estimation method based on the black box battery model usually requires the establishment of an offline database, and the sample training process has a large amount of calculation, which is prone to phenomena such as over-fitting and falling into local optimum. It is currently difficult to apply in engineering.

*3.3. SOC Estimation Method Based on the State Space Battery Model*

The state space model is based on the battery model to establish the system state space expression, and the battery SOC is used as one of the state variables, and then the battery SOC is estimated through the filter or observer [94]. The main idea is to link the measured current, voltage, temperature and other variables with the battery SOC. Taking these measurables as the input of the model, the error between the predicted value of the terminal voltage output by the model and the actual sampled value of the terminal voltage is obtained. Then multiply the error by the estimated value of the gain feedback to adjust the state quantity, so that the estimated value of the state quantity follows the true value. Finally, the current battery SOC value is obtained through the filter or the observer, as shown in Figure 6. The current research on the SOC estimation method based on the state space battery model mainly focuses on three aspects: the research on the battery equivalent circuit model structure, the research on the identification method of battery model parameters, and the research on the battery SOC estimation observer. The research on the structure of the equivalent circuit model was introduced in the summary of Section 2.2. The accuracy of the equivalent circuit model directly affects the accuracy of the SOC estimation based on the state-space battery model. Therefore, designing an equivalent circuit model with a simple structure and high accuracy will be the focus of research.

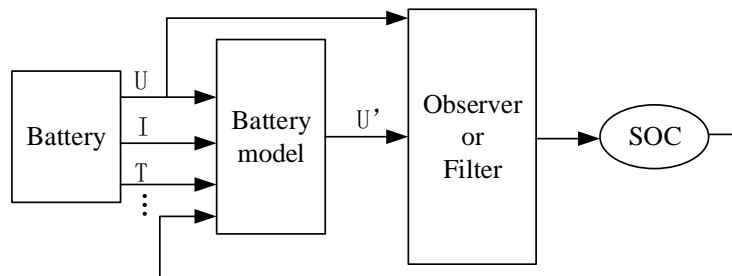

**Figure 6.** Structure diagram of SOC estimation method based on state space battery model.

3.3.1. Research on the Identification Method of Battery Model Parameters

The battery SOC estimation method based on the state space relies heavily on the accuracy of the battery model. The accuracy of model parameter identification directly affects the accuracy of the model output, thereby further affecting the accuracy of battery SOC estimation. At present, offline parameter identification and online parameter identification are the main research directions of battery model parameter identification methods.

The most commonly used parameter identification method is offline identification. This method first conducts a pulse test on the power battery. When the battery is switched from the charging or discharging state to the zero current output state, the voltage will have an instantaneous faster and a steady-state slow process. The instantaneous change is due to the ohmic internal resistance of the battery, while the slow change is due to the polarization characteristics of the battery. According to this feature, the parameters in the equivalent circuit model can be obtained by fitting the obtained experimental data using the least square method [95]. However, this method is easier to use for integer-order identification, while for fractional-order parameter identification, optimization algorithms

such as the particle swarm optimization algorithm and genetic algorithm need to be used for identification. The optimal solution is sought through iterative methods to reduce the offline identification error of model parameters [96,97]. However, since the operating conditions of the battery will change during actual use, if the offline parameter identification considers fewer external influence factors, it will cause a large identification error. Therefore, some scholars use the online parameter identification method. It is based on offline parameter identification, combines theoretical models and experimental data to increases the prediction technology of the consistency between the model output and the actual output. The recursive least squares method is the most widely used in the online identification of integer-order model parameters [98]. But the recursive least squares method has filter saturation problems in the actual use process, some scholars have proposed a recursive least squares method with forgetting factor and a decoupled weighted recursive least squares method [99]. However, this method ignores the influence of data with colored noise. Therefore, some scholars proposed the bias compensation recursive least squares (BCRLS) algorithm based on the recursive least squares with forgetting factor [100]. This method solves the identification problem of data with color noise by means of deviation compensation, and realizes unbiased identification of parameters. However, the input is required to have various states, which is difficult to achieve in the actual operation of the car. For the fractional model, it is a less computationally intensive and more accurate way to use the combined Kalman filter to identify the fractional order as a hidden variable. In summary, at present, further improving the real-time and accuracy of battery model parameter identification is the focus of research.

### 3.3.2. Research Status of SOC Estimation Observer

The working conditions of the car during the driving process are more complicated, the current changes more drastically, and the battery terminal voltage has strong non-linearity, which puts forward higher requirements on the battery SOC estimation observer. At present, most domestic and foreign experts and scholars have used various filtering methods based on Kalman filtering (KF) when performing SOC estimation. One of the most widely used is the use of extended Kalman filtering (EKF) to estimate the non-linear part of the battery [101–103]. EKF adopts the idea of minimum error to transform the nonlinear system into an approximate linear system, which has the advantage of overcoming the lack of sensor accuracy and correcting the initial value of SOC. But in the actual calculation process, the measurement noise and observation noise of the system change in real time. Therefore, some scholars have proposed adaptive extended Kalman filtering (AEKF) to estimate and iteratively update the noise covariance to reduce the impact of initial noise error on battery SOC estimation [104–106]. In addition to noise errors, when the current changes drastically, the observed voltage lag will cause a corresponding lag in the battery SOC estimation, which will also lead to a large deviation in the battery SOC estimation [95]. In order to solve this problem, reference [107] added a dynamic correction gain coefficient K to the EKF, and the gain coefficient can be dynamically adjusted when the current changes drastically to strengthen the algorithm convergence effect. Reference [108] uses the Levenberg-Marquardt method to modify the covariance matrix of EKF to ensure the convergence of the estimation process. In response to this situation, some scholars combine EKF with PID, robust control and other methods to reduce the estimation error when working conditions change drastically [109,110]. In order to solve the problem of linearization error caused by the use of EKF to estimate the battery SOC, the nonlinear function of the battery needs to be expanded into a Taylor series and the second-order and above terms are omitted. Reference [111] uses a proportional-integral correction method to compensate for the error generated in the EKF linearization model. Since this error cannot be completely eliminated, some scholars have used an improved form based on Kalman filtering without a linearized model, such as cubature Kalman filtering (CKF) [112], and unscented Kalman filtering (UKF) [73,74] to estimate the battery SOC. In addition to Kalman filtering, other forms of observers were also extensively studied by domestic

and foreign experts, such as the statistical filter algorithm [113], improved particle filter algorithm [114–116], linear inequality estimation method based on H∞ [117], sliding mode observer [118,119], proportional integral observer [120], Luenberger observation [121,122], and so on. They have achieved good estimation accuracy and convergence in battery SOC estimation. Both the observer algorithm and the Kalman filter algorithm mentioned above need to establish a battery model, and then estimate the power battery SOC as a state quantity. However, the amount of calculation is smaller than that of Kalman filter, and it has strong robustness to nonlinear systems, which improves the adaptability of the algorithm.

## 4. Summary

The second chapter of the article gives a comprehensive overview of the electrochemical mechanism model, equivalent circuit model and data-driven model. According to research and analysis, each battery model has its own advantages and disadvantages, as shown in Table 2. The accuracy of SOC estimation depends on the accuracy of the model. With higher model accuracy, the model will become relatively complicated. The electrochemical mechanism model can better reflect the internal chemical reaction principle of the battery, and the estimation accuracy is higher. However, the amount of calculation is large, the calculation is complicated, and dimensionality reduction processing is required, which takes a long time. In current research, the improved P2D and SP2D models are often used in conjunction with other intelligent algorithms to estimate the battery SOC, which can maintain the accuracy of the estimation while reducing the amount of calculation. The equivalent circuit model simulates the external characteristics of the battery with ideal electrical components. Its estimation accuracy is worse than that of the electrochemical mechanism model, but its structure is simple, and the parameters are easy to obtain, which is suitable for battery management systems. The data-driven model eliminates the tedious modeling process due to its model-free advantage and can quickly evaluate and analyze the internal state of the battery. However, it has a high dependence on the number of samples and a slower convergence speed.

According to the choice of battery model, the third chapter of the article systematically sorts out the direct detection method that is not based on the battery model, the estimation method based on the black box battery model, and the estimation method based on the shape space. The pros and cons of the respective estimation methods are shown in Table 3. For direct detection methods that are not based on the battery model, the open circuit voltage method has the highest estimation accuracy, but because it requires a long time to stand, it is difficult to apply to actual vehicle use. The ampere-hour integration method is also widely used because of its simplicity and reliability, and its low requirements for equipment. However, it depends very much on the accuracy of the initial value and the accuracy of the sensor measurement value. After a long time of current integration, there will be accumulated errors. Internal resistance method, discharge test method, and electrochemical impedance method estimation method all have high estimation accuracy, but the requirements for equipment and test conditions are high. The load voltage method is greatly affected by the circuit and is not suitable for practical applications. The SOC estimation method based on the black box battery model has strong learning ability and high estimation accuracy. The neural network is based on the existing data. The more data there is, the higher the estimation accuracy. Support vector machines have good generalization ability and nonlinear approximation ability, which can effectively avoid the shortcomings of neural networks. And deep learning has higher estimation accuracy and stability. As it can only deal with short timing problems, it can be improved through LSTM and GRU.

**Table 2.** Battery model comparison.

| Type of model | Electrochemical mechanism model | Equivalent circuit model | Data-driven models |
|---|---|---|---|
| Accuracy | Very high | Medium | Medium |
| Computational Complexity | Very high | Medium to low | Medium |
| Configuration Effort | high | Medium | Medium to high |
| Time | Solving control equations consumes a lot of time | Simple and easily understood, so medium time consuming | Less time consuming as prior battery knowledge is not required |
| Interpret Ability | Low | High | Low |
| Merits | The mathematical model established by the knowledge of electrochemical theory can better reflect the characteristics of the battery and have Very high accuracy | Simple structure. Easy access to model parameters | Do not rely on the battery model, eliminating the tedious process of physical modeling. Can quickly evaluate and analyze the internal state of the battery |
| Demerits | Poor adaptability to some working conditions, leading to poor estimation results | Can not reflect the internal characteristics of the battery well | The estimation accuracy depends heavily on the number of samples, and the convergence speed is slow. When the sample size is small and the numerical error rate is high, the model will be over-fitted and under-fitted |

**Table 3.** Comparison of SOC estimation methods.

| Estimation Method | | Merit | Demerit |
|---|---|---|---|
| Direct measurement method not based on battery model | Ampere-hour integral method | Simple and reliable, fast estimation speed, low requirements for controller hardware and storage | The sensor has high requirements for accuracy, which is heavily dependent on the accuracy of the initial SOC value, and there is a cumulative error |
| | Open circuit voltage method | Simple structure, convenient operation and high estimation accuracy | Long standing time and hysteresis effect |
| | Internal resistance method | The principle is simple, and the estimation accuracy is high | The resistance test device is expensive, the internal resistance value is small, the range of change is small, and the resistance is easily affected by the temperature and the number of cycles |
| | Discharge test method | High estimation accuracy and strong reliability | It takes a long time and requires high test conditions, and it is impossible to estimate the battery SOC value in real time |
| | Electrochemical impedance spectroscopy | High estimation accuracy, which can better reflect the dynamic characteristics of the battery | High battery impedance cost, susceptible to battery temperature and life |
| | Load voltage method | Good estimation accuracy under constant current conditions | Affected by current changes, it is not suitable for practical applications |

**Table 3.** *Cont.*

| Estimation Method | | Merit | Demerit |
|---|---|---|---|
| Estimation method based on black box model | Neural Networks | No battery model is required, with strong variable processing ability and self-learning ability, real-time detection of SOC status | Severely depends on the number of samples, and samples have a greater impact on the training results, long learning time, and heavy sampling workload |
| | Support Vector Machines | It has strong generalization ability, does not rely on the battery model, and has high estimation accuracy and fast convergence speed in the case of small samples | The estimation accuracy depends heavily on a large number of sample data and weight parameters |
| | Deep learning | It has strong generalization ability and parallel processing ability, and the estimation result has high accuracy and stability | Model training is complex, requires high computing resources and configuration, and has over-fitting problems |
| | Genetic algorithm | Highly parallel operation, self-organization, self-adaptation, self-learning and group evolution capabilities, high robustness | The algorithm is complex and the global search speed is slow, and it is easy to fall into the local optimum |
| State space-based estimation method | Kalman filter | The estimation accuracy is high in the case of considering the error, does not depend on the initial SOC value, and has a strong anti-interference ability | The estimation accuracy depends on the accuracy of the model, is easily affected by temperature, and is limited to linear systems |
| | Extended Kalman filter | Suitable for non-linear systems, suitable for working conditions with severe current fluctuations | Ignoring high-order terms in the linearization process produces a large error value and poor robustness |
| | Double Kalman filter | High estimation accuracy, which can effectively eliminate noise in the system and model | The amount of calculation is large, and the calculation takes a long time |
| | Unscented Kalman Filtering Method | Suitable for nonlinear systems, reducing errors caused by linear systems | Factors such as abnormal disturbance and initial value uncertainty cause the system to diverge, and its robustness is poor |
| | Adaptive Kalman filter | Able to continuously estimate the system status in real time and correct the influence of noise | Need noise zero mean hypothesis and noise variance is known, and the measured value may diverge |
| | Particle filter | It is not restricted by the linear and Gaussian conditions of the system model, and has few constraints on the probability distribution of state variables | The estimation accuracy is not stable, and the phenomenon of particle depletion is prone to occur |

## 5. Future Development

The choice of model mainly depends on environmental conditions, operating temperature, battery aging, application scenarios and different SOC operating ranges. According to the analysis of selection factors for battery model selection, the accuracy of model performance can be improved through the following aspects: OCV and hysteresis are modeled as a function of temperature and SOC; the model should have good accuracy and adaptability to accurately describe the battery characteristics under multiple conditions and multiple states; the model can better reflect the dynamic and static characteristics of the battery, and the number of model components should be reduced to reduce the amount of model calculations and improve the application of model engineering; aging effects are included in the model.

The research hotspots and development trends of future battery SOC estimation methods need to focus on the following aspects: an accurate battery model is a prerequisite for achieving high-precision estimation, so a battery model with excellent accuracy and complexity should be further developed; multi-constrained SOC estimation of battery internal



resistance, ambient temperature, battery aging state, and discharge rate can be considered to improve the accuracy of algorithm estimation; because SOC estimation methods have their own unique advantages, a variety of methods are used to comprehensively complement each other to further improve the estimation accuracy; considering the problem of cost estimation, the algorithm development cycle should be shortened as much as possible while achieving low cost.

**Author Contributions:** All of the authors contributed to publishing this article. All authors have carefully studied the relevant literature on power battery models and battery SOC estimation research. W.Z. is mainly responsible for sorting out the overall thinking of the paper, writing the paper and subsequent revisions. Z.P. is mainly responsible for sorting out the materials needed for the thesis and cooperating with the writing of W.Z. thesis. Q.L. is responsible for paper inspection and typesetting. Y.Z. provides technical guidance during the writing and revision of the paper. All authors have read and agreed to the published version of the manuscript.

**Funding:** We gratefully acknowledge the financial support of the National Natural Science Foundation of China (No. 51306079 and 51176069) and the Jiangsu Provincial Key Research and Development Program (No. BE2017008).

**Conflicts of Interest:** The authors declare no conflict of interest.

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
