# Peer review of "Review on the Battery Model and SOC Estimation Method"

_processes, doi:10.3390/pr9091685_

Round 1

Reviewer 1 Report

  1. Part of the introduction is missing. The introduction details the importance and challenges of accurate SOC estimation. However, the subject of this paper includes the battery model and SOC estimation method. Therefore, the introduction lacks the content of the battery model.
  1. There are no references in the introduction.
  2. Some sentences are too long and have language problems. For example, the last sentence of the introduction is too long. The subject of “analyze” is not clear.
  1. The number of charts is small, and only two pictures belong to the same section.
  2. The description of figure 1 is unreasonable. Figures 1 and 2 show common integer-order models and fractional-order models, respectively. The description of figure 1, “Common equivalent circuit model”, is unreasonable.

Reviewer 2 Report

The authors made a review on battery model and SOC estimation methods, which is comprehensive. However, there are still some problems need to be address.

  1. Could you explain your survey methodology like how you get and screen the articles cited in your review?
  2. A bit more details are suggested to be given about polarization, aging and individual differences that appeared in line 50 to 52.
  3. Title of section 3.1 is vague, and you are supposed to change it to a more appropriate one.
  4. Please give a direct definition about black box model and state space model in the front of section 3.2 and section 3.3.
  5. Some essential tables about the mentioned methods’ key information, such as error, complexity, are supposed to be given.
  6. The prospect proposed in section 4 needs a more detailed analysis.
  7. As long-short term memory and gated recurrent unit is widely used, why did not the authors mention these algorithms.

Reviewer 3 Report

The authors have reviewed the SOC-estimation methods. The reviewer has the following concerns:

  1. There are already several review papers that have already reviewed the SOC estimation methods. The reviewer could not find any new point in the paper.
  2. In the introduction section, several published articles have been discussed quite briefly. It does not give a clear idea and complete picture of the research work to the reader. Therefore, the reviewer suggests a table of a detailed comparison of the published work with the pros and cons of the work should be given in the introduction section.
  3. Similarly, a table of techniques should be provided which must give a complete picture of all techniques currently utilized for SOC estimation.
  4. Also, provide a comparison, after extensive reviewing of the literature what are the findings of the author? Which method is best among all SOC estimation methods?
  5. The future work section is missing. Include a separate sub-section and discuss all the possibilities of future research.

Round 2

Reviewer 3 Report

The authors have addressed my comments.